# A deleterious gene-by-environment interaction imposed by calcium channel blockers in Marfan syndrome

Jefferson J Doyle[1,2]*[†], Alexander J Doyle[1,3][†], Nicole K Wilson[1], Jennifer P Habashi[1,4], Djahida Bedja[5,6], Ryan E Whitworth[7], Mark E Lindsay[8], Florian Schoenhoff[9,10], Loretha Myers[1], Nick Huso[1], Suha Bachir[1], Oliver Squires[1], Benjamin Rusholme[1], Hamid Ehsan[2], David Huso[11], Craig J Thomas[12], Mark J Caulfield[3], Jennifer E Van Eyk[10], Daniel P Judge[13], Harry C Dietz[1,4,13]*, GenTAC Registry Consortium, MIBAVA Leducq Consortium

[1]Howard Hughes Medical Institute and Institute of Genetic Medicine, Johns Hopkins University School of Medicine, Baltimore, United States; [2]Wilmer Eye Institute, Johns Hopkins University School of Medicine, Baltimore, United States; [3]William Harvey Research Institute, Barts and The London School of Medicine, Queen Mary University of London, London, United Kingdom; [4]Division of Pediatric Cardiology, Department of Pediatrics, Johns Hopkins University School of Medicine, Baltimore, United States; [5]Department of Cardiology, Johns Hopkins University School of Medicine, Baltimore, United States; [6]Australian School of Advanced Medicine, Macquarie University, Sydney, Australia; [7]Research Triangle Institute International, Durham, United States; [8]Thoracic Aortic Center, Departments of Medicine and Pediatrics, Massachusetts General Hospital, Harvard Medical School, Boston, United States; [9]Department of Cardiovascular Surgery, Inselspital, Bern, Switzerland; [10]Proteomics Innovation Center in Heart Failure, Johns Hopkins University School of Medicine, Baltimore, United States; [11]Department of Molecular and Comparative Pathobiology, Johns Hopkins University School of Medicine, Baltimore, United States; [12]Division of Preclinical Innovation, National Center for Advancing Translational Sciences, Bethesda, United States; [13]Department of Medicine, Johns Hopkins University School of Medicine, Baltimore, United States

*For correspondence: jefferson. doyle@jhmi.edu (JJD); hdietz@ jhmi.edu (HCD)

[†]These authors contributed equally to this work

Group author details
GenTAC Registry Consortium
Group author details
MIBAVA Leducq Consortium
See page 15

Competing interests:
See page 15

**Abstract** Calcium channel blockers (CCBs) are prescribed to patients with Marfan syndrome for prophylaxis against aortic aneurysm progression, despite limited evidence for their efficacy and safety in the disorder. Unexpectedly, Marfan mice treated with CCBs show accelerated aneurysm expansion, rupture, and premature lethality. This effect is both extracellular signal-regulated kinase (ERK1/2) dependent and angiotensin-II type 1 receptor (AT1R) dependent. We have identified protein kinase C beta (PKCβ) as a critical mediator of this pathway and demonstrate that the PKCβ inhibitor enzastaurin, and the clinically available anti-hypertensive agent hydralazine, both normalize aortic growth in Marfan mice, in association with reduced PKCβ and ERK1/2 activation. Furthermore, patients with Marfan syndrome and other forms of inherited thoracic aortic aneurysm taking CCBs display increased risk of aortic dissection and need for aortic surgery, compared to patients on other antihypertensive agents.

## Introduction

Marfan syndrome is a systemic connective tissue disorder caused by mutations in *FBN1*, the gene encoding extracellular matrix protein fibrillin-1. A major cause of mortality in Marfan patients is aortic dissection and rupture. In Marfan mice, multiple phenotypic manifestations, including aortic

**eLife digest** Marfan syndrome is a disorder that affects the body's connective tissues, which maintain the structure of the body and support organs and other tissues. People with Marfan syndrome have connective tissues that can stretch more than those of other people, which put them at increased risk of a life-threatening tear in their aorta (the main artery in the body), muscle weakness and other problems.

A cell communication pathway called TGFβ signaling is involved in cell growth and many other important processes. TGFβ signaling is more active in patients with Marfan syndrome due to mutations in a gene called *FBN1*. Drugs that block TGFβ signaling—which are also used to treat high blood pressure—can reduce the symptoms of the disorder. Unfortunately, not all people with Marfan disease can tolerate these drugs and other medications called calcium channel blockers, which also lower blood pressure, are often used as an alternative. It is thought that calcium channel blockers help reduce stress on blood vessels, but there is little data to show whether these drugs are safe and helpful for patients with Marfan syndrome.

Now, Doyle, Doyle et al. studied the effect of two different calcium channel blockers on mice that have a mutation in *Fbn1*—the mouse equivalent of *FBN1*—that is similar to those found in humans with Marfan syndrome. The experiments show that the aortas of these mice grew more quickly and were more likely to tear when compared to mice that did not receive these drugs. Many of these aortic tears were fatal. The calcium channel blockers increased the activity of two signaling molecules that are regulated by TGFβ signaling. Treating the Marfan mice with other drugs that lower the activity of these signaling molecules protected the aorta, even if they were also treated with the calcium channel blockers.

Doyle, Doyle et al. examined a registry of human patients. This revealed preliminary evidence that aortic tears and aortic repair surgery were more common in patients with Marfan syndrome who had received calcium channel blockers than patients who had been treated with other drugs. Together, these findings suggest that it may be dangerous to treat patients with Marfan syndrome with calcium channel blockers. Additional work will be needed to confirm this risk, to find out if it extends to other similar conditions, and to explore the therapeutic potential of drugs that target the two enzymes.

aneurysm, developmental lung emphysema, mitral valve disease, and skeletal muscle myopathy, correlate with enhanced transforming growth factor beta (TGFβ) signaling, while treatment with either TGFβ neutralizing antibody (NAb) or the angiotensin-II type 1 receptor (AT1R) blocker (ARB) losartan can ameliorate these phenotypes, in association with evidence of reduced TGFβ signaling (*Neptune et al., 2003*; *Ng et al., 2004*; *Habashi et al., 2006*; *Cohn et al., 2007*; *Cook et al., 2015*). Both canonical (Smad2/3) and noncanonical (ERK1/2) TGFβ-dependent signaling cascades have been shown to be activated in the aortas of Marfan mice, while selective inhibition of extracellular signal-regulated kinase (ERK1/2) activation using RDEA119 (refametinib) rescues aortic growth and aortic wall architecture in Marfan mice (*Habashi et al., 2011*; *Holm et al., 2011*).

Calcium channel blockers (CCBs) are a class of blood pressure lowering medications that block movement of calcium into cells from the extracellular space. There are several sub-classes of CCB based on chemical structure, including dihydropyridines (e.g., amlodipine) and non-dihydropyridines, which include both phenylalkylamines (e.g., verapamil) and benzothiazepines (e.g., diltiazem). CCBs are currently considered an alternative therapeutic strategy for Marfan patients intolerant of β-blockers, due to their ability to reduce contractility of the heart (negative ionotropy) and to lower blood pressure (*Milewicz et al., 2005*; *von Kodolitsch and Robinson, 2007*; *Hartog et al., 2012*). In doing so, they theoretically reduce stress on the aortic wall during cardiac systole. However, there is limited empirical evidence for their safety or utility in patients with Marfan syndrome and other forms of inherited thoracic aortic aneurysm (*Williams et al., 2008*). To address this, we assessed the effect of two major classes of CCB in a mouse line heterozygous for a cysteine substitution in an epidermal growth factor-like domain in fibrillin-1 ($Fbn1^{C1039G/+}$), representative of the most common class of mutations causing Marfan syndrome. This mouse model has been shown previously to closely recapitulate many of the phenotypic manifestations seen in patients with the disorder, including aortic aneurysm (*Habashi et al., 2006*; *Holm et al., 2011*; *Cook et al., 2015*).

## Results

### Effect of dihydropyridine CCBs: amlodipine

Wild-type (WT) and Marfan mice were treated with either placebo or amlodipine from 2 to 5 months of age. They were treated with a dose of amlodipine (12 mg/kg/day) that resulted in a similar reduction in blood pressure as is obtained using losartan in our Marfan mouse model (*Figure 1—figure supplement 1S1*). They underwent unsedated in vivo echocardiography at baseline prior to treatment and then every month thereafter. Between 2 and 4 months of age, placebo-treated Marfan mice showed greater aortic root growth than WT littermates, which was unexpectedly exacerbated by amlodipine. This exacerbation was seen in both WT and Marfan mice but was of greater magnitude in the Marfan animals, indicating a specific interaction between the drug and the genotype of the mice (genotype effect: p < 0.0001, amlodipine effect: p < 0.001, interaction effect: p < 0.01, *Figure 1A*). Amlodipine-treated WT and Marfan mice also showed striking dilatation of the ascending aorta, an aortic segment just distal to the aortic root, which is less commonly affected in people and mice with Marfan syndrome. Amlodipine-induced growth in the ascending aorta was several fold greater than that seen in the aortic root, with Marfan mice again appearing particularly susceptible (genotype effect: p < 0.0001, amlodipine effect: p < 0.0001, interaction effect: p < 0.0001, *Figure 1A*). Furthermore, while no WT or placebo-treated Marfan mice died during the 3-month drug trial, more than 40% of amlodipine-treated Marfan mice died secondary to aortic rupture, as evidenced by hemothorax or hemopericardium (p < 0.01, *Figure 1B*).

Following death or sacrifice, latex injection of the vasculature illustrated the enlargement of the ascending aorta in amlodipine-treated Marfan mice (*Figure 1—figure supplement 1S2*). Histologic analyses revealed greater ascending aortic wall thickening, elastic fiber fragmentation, reduced elastin content, and increased collagen deposition in Marfan animals compared to WT littermates, all of which were exacerbated by amlodipine treatment (*Figure 1C*). Aortic architecture was graded quantitatively on a scale of 0–5, by 4 observers blinded to both genotype and treatment arm, as described previously (*Holm et al., 2011*; *Cook et al., 2015*). While amlodipine had no effect on aortic architecture in WT animals (p = 0.16), it induced significant histological damage in Marfan mice (interaction effect: p < 0.001, *Figure 1D*). Interestingly, these deleterious histological effects of amlodipine were not observed in the descending thoracic aorta of Marfan mice (*Figure 1—figure supplement 1S3*; p = 0.86, *Figure 1—figure supplement 1S4*), an aortic segment that lacks predisposition for dilatation or evidence of increased TGFβ signaling in Marfan mice (*Haskett et al., 2012*).

Prior studies assessing the effects of CCBs in other mouse models of aortic disease, such as the angiotensin-II infusion model, have typically used lower doses of amlodipine, in the range of 1 to 5 mg/kg/day (*Chen et al., 2013*; *Takahashi et al., 2013*). We utilized a dose of 12 mg/kg/day in order to obtain a similar reduction in blood pressure as is achieved using losartan in our Marfan mouse model, as we wished to eliminate blood pressure as a confounding variable when comparing the differential physiological and biochemical effects of the two agents. To ensure that the deleterious effect we observed was not simply due to toxicity of the drug, we repeated the trial using a dose of 3 mg/kg/day. Even at this lower dose, amlodipine still exacerbated both aortic root and ascending aortic growth in WT mice and Marfan littermates, with greater accentuation again occurring in the ascending aorta and in Marfan animals (aortic root interaction effect: p < 0.01, ascending aorta interaction effect: p < 0.01, *Figure 1—figure supplement 1S5*).

### Effect of non-dihydropyridine CCBs: verapamil

To assess the generalizability of our observations with amlodipine, we treated WT and Marfan mice with verapamil from 2 to 6 months of age. Compared to placebo-treated WT and Marfan mice, verapamil-treated animals showed enhanced growth in both the aortic root and ascending aorta, with greater accentuation again occurring in Marfan animals, indicating a specific interaction between the drug and the genotype of the mice (aortic root interaction effect: p < 0.01, ascending aorta interaction effect: p < 0.01, *Figure 2A*; *Figure 2—figure supplement 1S1*). While verapamil had no effect on aortic architecture in WT mice, it significantly exacerbated the histological changes seen in the ascending aortic wall of Marfan mice, both qualitatively (*Figure 2B*) and quantitatively (interaction effect: p < 0.0001, *Figure 2C*). As with amlodipine, verapamil had no effect on aortic architecture in

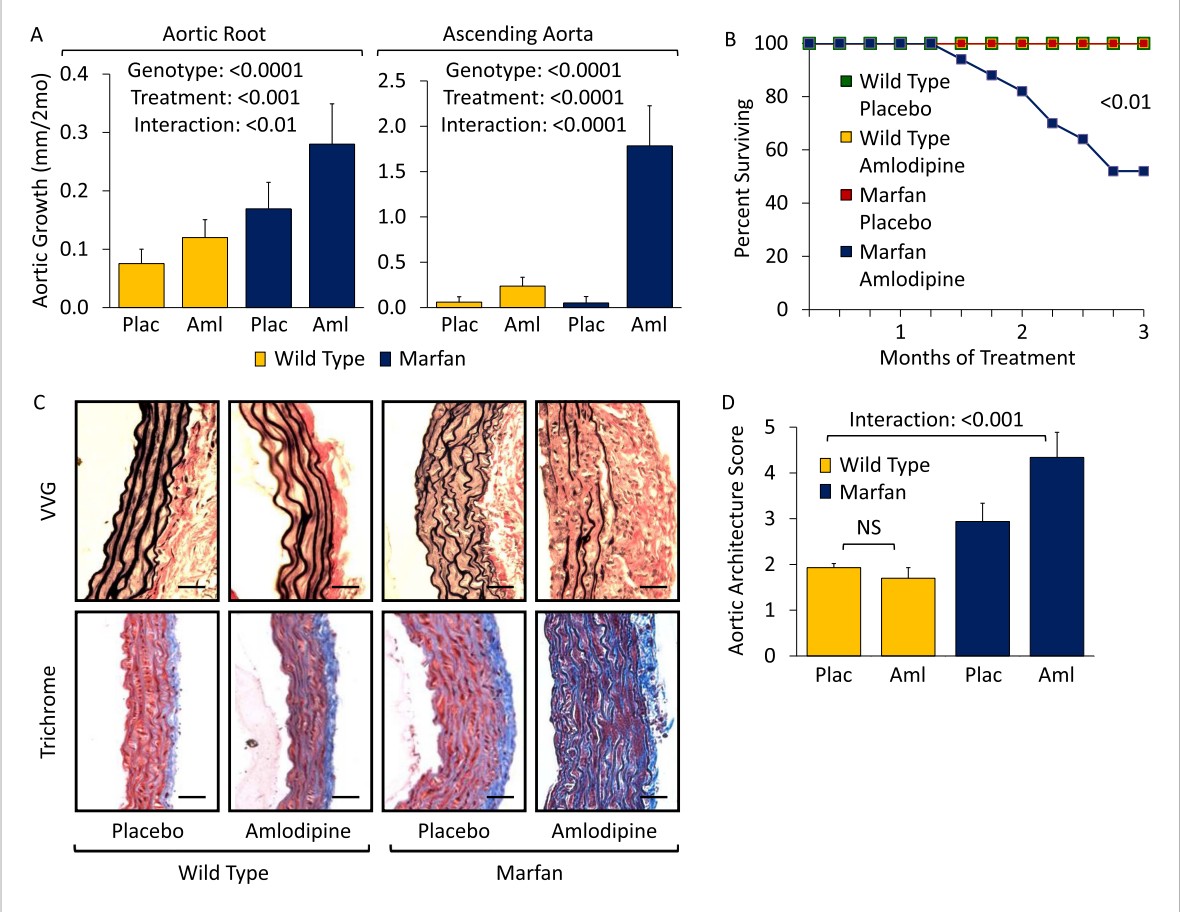

Figure 1. Effect of amlodipine in wild-type (WT) and Marfan mice. (A) Echocardiography data showing mean (±2 SEM) growth in the aortic root and ascending aorta from 2 to 4 months of age. Number of mice per group (male/female) = WT placebo 11 (5/6), WT amlodipine 10 (6/4), Marfan placebo 14 (7/7), Marfan amlodipine 11 (6/5). Mean (±2 SEM) weight per group (in grams) at 4 months = 27.4 ± 2.4 g (WT placebo), 27.6 ± 2.8 g (WT amlodipine), 28.1 ± 3.0 g (Marfan placebo), 27.9 ± 2.8 g (Marfan amlodipine). (B) Survival curve from 2 to 5 months of age. Number of mice per group (male/female) = WT placebo 11 (5/6), WT amlodipine 10 (6/4), Marfan placebo 14 (7/7), Marfan amlodipine 19 (9/10). (C) Representative VVG staining (upper panel) and Masson's trichrome staining (lower panel) of the proximal ascending aorta in 5-month-old male mice. Scale bar: 40 μm. (D) Mean (±2 SEM) aortic wall architecture score of the proximal ascending aorta in 5-month-old mice. Number of mice per group = 4 (2 male, 2 female). Scale: 1 (normal) to 5 (extensive damage). Plac, placebo; Aml, amlodipine.

The following figure supplement is available for figure 1:

Figure supplement 1. Effect of amlodipine in wild-type (WT) and Marfan mice.

the descending thoracic aorta of Marfan mice (*Figure 2—figure supplement 1S2*; p = 0.72, *Figure 2—figure supplement 1S3*).

## Mechanism: a role for AT1R-mediated ERK1/2 activation

To interrogate the mechanism underlying this detrimental CCB effect in Marfan mice, we performed Western blot analyses on the aortas of 5-month-old animals (*Figure 3A*). Treatment with amlodipine accentuated signaling changes previously observed in Marfan mice, including enhanced activation of both canonical (Smad) and noncanonical (ERK1/2) TGFβ-dependent signaling cascades, when normalized to either β-Actin or their respective total proteins (Smad3 treatment effect: p < 0.0001, ERK1/2 treatment effect: p < 0.01, *Figure 3A*). Again, amlodipine had a greater effect in Marfan mice than WT littermates, indicating a specific interaction between the drug and the genotype of the animals (Smad3 interaction effect: p < 0.001, ERK1/2 interaction effect: p < 0.01, *Figure 3A*).

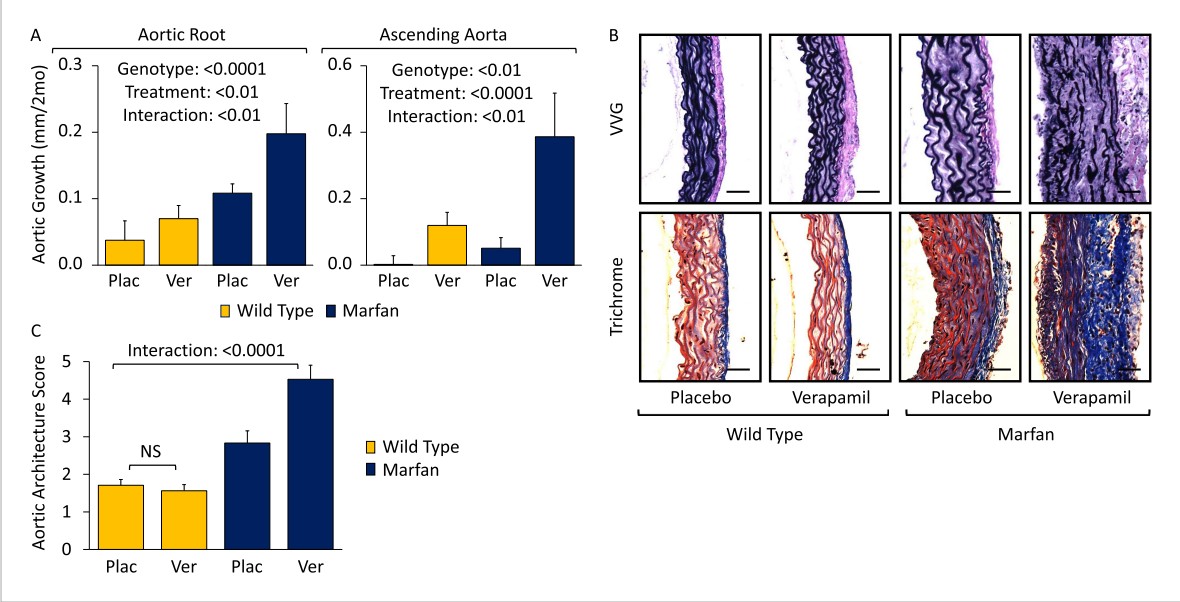

**Figure 2**. Effect of verapamil in wild-type (WT) and Marfan mice. (**A**) Mean (±2 SEM) growth in the aortic root and ascending aorta from 2 to 6 months of age. Number of mice per group (male/female) = WT placebo 8 (4/4), WT verapamil 10 (4/6), Marfan placebo 9 (5/4), Marfan verapamil 11 (6/5). Mean (±2 SEM) weight per group (in grams) at 6 months = 30.3 ± 2.6 g (WT placebo), 30.6 ± 2.2 g (WT verapamil), 31.3 ± 3.3 g (Marfan placebo), 31.5 ± 3.2 g (Marfan verapamil). (**B**) Representative VVG staining (upper panel) and Masson's trichrome staining (lower panel) of the proximal ascending aorta in 6-month-old male mice. Scale bar: 40 μm. (**C**) Mean (±2 SEM) aortic wall architecture score of the proximal ascending aorta in 6-month-old mice. Number of mice per group = 4 (2 male, 2 female). Scale: 1 (normal) to 5 (extensive damage). Plac, placebo; Ver, verapamil.

The following figure supplement is available for figure 2:

**Figure supplement 1**. Effect of verapamil in wild-type (WT) and Marfan mice.

Combined treatment with amlodipine and the selective inhibitor of ERK1/2 activation RDEA119 prevented amlodipine-induced ascending aortic enlargement (treatment effect: p < 0.0001), with the magnitude of effect being significantly greater in Marfan mice (interaction effect: p < 0.001, *Figure 3B*). Furthermore, RDEA119 rescued the premature lethality (p < 0.05, *Figure 3C*), and alterations in aortic wall architecture (*Figure 3—figure supplement 1S1*; interaction effect: p < 0.0001, *Figure 3—figure supplement 1S2*), seen in amlodipine-treated Marfan mice, in association with abrogated ERK1/2 activation (treatment effect: p < 0.0001, *Figure 3D*). RDEA119 had no significant effect on Smad3 activation (p = 0.26, *Figure 3D*), suggesting that rescue of CCB-induced aortic aneurysm exacerbation in Marfan mice can occur independent of Smad3 activation status. In contrast, these data suggest that enhanced ERK1/2 activation is a critical mediator of CCB-mediated aortic aneurysm progression in Marfan mice.

ERK1/2 activation in Marfan mice has previously been shown to also be AT1R dependent, since the AT1R blocker (ARB) losartan can rescue aortic root growth in Marfan animals in association with normalization of ERK1/2 activation (*Holm et al., 2011*). Losartan also prevented amlodipine-induced ascending aortic aneurysm growth (treatment effect: p < 0.01), with the magnitude of effect being significantly greater in Marfan mice (interaction effect: p < 0.001, *Figure 3E*). This correlated with reduced ERK1/2 activation in these animals (treatment effect: p < 0.0001, *Figure 3F*). Hence the deleterious effect of CCBs in Marfan mice also appears to be AT1R-dependent.

## Mechanism: a role for PKC activation

There is evidence that TGFβ-dependent gene expression can be mediated by protein kinase C (PKC) in both human fibroblasts and aortic vascular smooth muscle cells (VSMCs) (*Mulsow et al., 2005*; *Ryer et al., 2006*), and PKC blockade can inhibit TGFβ-dependent phenotypes and gene expression in vitro and in vivo (*Li and Jimenez, 2011*; *Lee et al., 2013*). Furthermore, angiotensin-II has been shown to

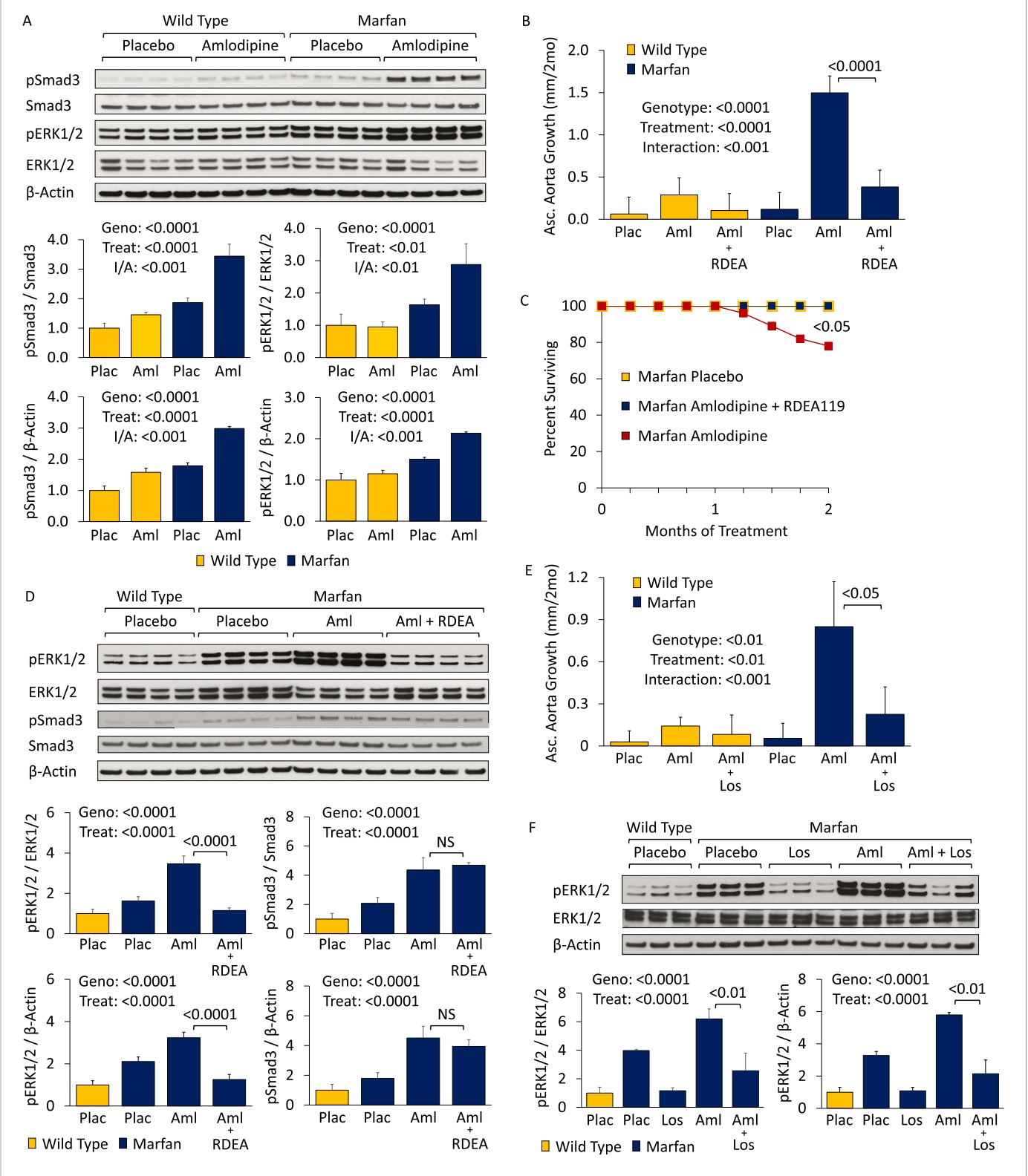

**Figure 3**. CCB effect is ERK1/2- and AT1R-dependent in wild-type (WT) and Marfan mice. (**A**) Western blot analysis of the aortic root and ascending aorta in 5-month-old mice. Number of mice per group = 4 (2 male, 2 female). (**B**) Mean (±2 SEM) ascending aortic growth from 2 to 4 months of age. Number of mice per group (male/female) = WT placebo 9 (5/4), WT amlodipine 8 (4/4), WT amlodipine + RDEA119 7 (3/4), Marfan placebo 9 (4/5), Marfan amlodipine 10 (6/4), Marfan amlodipine + RDEA119 11 (6/5). (**C**) Survival curve from 2 to 4 months of age. (**D**) Western blot analysis of the aortic root and ascending
*Figure 3. continued on next page*

Figure 3. Continued

aorta in 4-month-old mice. Number of mice per group = 4 (2 male, 2 female). (**E**) Mean (±2 SEM) ascending aortic growth from 2 to 4 months of age. Number of mice per group (male/female) = WT placebo 11 (6/5), WT amlodipine 11 (6/5), WT amlodipine + losartan 8 (4/4), Marfan placebo 7 (4/3), Marfan amlodipine 6 (3/3), Marfan amlodipine + losartan 9 (4/5). (**F**) Western blot analysis of the aortic root and ascending aorta in 4-month-old mice. Number of mice per group = 3 (2 male, 1 female; or 1 male, 2 female). Plac, placebo; Aml, amlodipine; RDEA, RDEA119; Los, losartan; Geno, genotype; Treat, treatment; I/A, interaction.

The following figure supplement is available for figure 3:

**Figure supplement 1**. CCB effect is ERK1/2- and AT1R-dependent in wild-type (WT) and Marfan mice.

activate ERK1/2 via PKC (*Shah and Catt, 2002*; *Olivares Reyes et al., 2005*; *Olson et al., 2008*), with TGFβ potentially serving as an intermediary in this process (*Uchiyama-Tanaka et al., 2001*).

Western blot analysis of the aortic root and ascending aorta showed that PKCβ activation was significantly greater in Marfan mice than WT littermates (genotype effect: $p < 0.0001$), and both TGFβ NAb and losartan significantly reduced it (treatment effect: $p < 0.0001$ for both, *Figure 4A,B*). This closely paralleled their effects on ERK1/2 activation (*Figures 3F, 4A*). Furthermore, amlodipine treatment accentuated PKCβ activation in Marfan mice, in association with increased activation of phospholipase C (PLCγ), its upstream activator, both of which were rescued by treatment with losartan (treatment effect: $p < 0.0001$ for both, *Figure 4B*). These data suggest that PKCβ and its upstream activator PLCγ are both TGFβ- and AT1R-dependent in Marfan mice.

Enzastaurin is a PKC inhibitor with relative selectivity for PKCβ and has been shown previously to inhibit PKC-mediated ERK1/2 activation (*Ruvolo et al., 2011*; *Wu et al., 2012*). It competes with ATP for the nucleotide triphosphate-binding site of PKC, thereby blocking its activation (*Graff et al., 2005*). In keeping with the hypothesis that PKCβ mediates CCB-induced aortic aneurysm exacerbation in Marfan mice, treatment with enzastaurin led to a significant reduction in ascending aortic growth in amlodipine-treated animals (treatment effect: $p < 0.0001$), with the magnitude of effect being significantly greater in Marfan mice (interaction effect: $p < 0.0001$, *Figure 4C*). Latex-injected images show the rescue achieved by enzastaurin on ascending aortic aneurysm in amlodipine-treated Marfan mice (*Figure 4—figure supplement 1S1*). Enzastaurin also significantly rescued the deleterious histological changes imposed on the Marfan aorta by amlodipine, both qualitatively (*Figure 4—figure supplement 1S2*) and quantitatively (treatment effect: $p < 0.0001$, *Figure 4—figure supplement 1S3*). Cumulatively, these data suggest that PKCβ mediates amlodipine-induced aortic aneurysm exacerbation in Marfan mice.

To confirm that PKCβ also mediates aortic aneurysm progression in placebo-treated Marfan mice, aortic root growth was measured in WT and Marfan animals over a 2-month treatment period (*Figure 4D*). This showed that enzastaurin was indeed able to significantly reduce aortic root growth in both WT and Marfan mice (treatment effect: $p < 0.0001$). While the magnitude of effect was greater in Marfan mice than WT littermates, this difference trended towards significance but did not quite reach it (interaction effect: $p = 0.09$).

Western blot analysis of the aortic root and ascending aorta confirmed that the protection conferred on aortic growth by enzastaurin correlated with a significant reduction in PKCβ activation in both placebo- and amlodipine-treated Marfan mice (treatment effect: $p < 0.0001$, *Figure 4E*). Furthermore, enzastaurin was able to significantly rescue ERK1/2 activation in these animals (treatment effect: $p < 0.0001$, *Figure 4E*), inferring that PKCβ may mediate ERK1/2 activation in this setting.

## Hydralazine: a novel therapeutic strategy in Marfan syndrome

We became interested in the clinically available antihypertensive agent hydralazine, not only because it reduces blood pressure (a desirable effect in Marfan syndrome) but also because it has been shown to inhibit PKC-mediated ERK1/2 activation in vivo (*Deng et al., 2003*; *Gorelik et al., 2007*). Therefore, we performed a trial of WT and Marfan mice treated with hydralazine from 2 to 6 months of age, at a dose of 16 mg/kg/day (*Doblinger et al., 2012*; *Shinmura et al., 2015*). This dose of hydralazine reduced systolic and diastolic blood pressure by roughly 10–15% in our mice (*Figure 5—figure supplement 1S1*). Compared to WT mice, placebo-treated Marfan animals showed greater aortic root growth, which was fully rescued by hydralazine (genotype effect: $p < 0.001$, treatment effect:

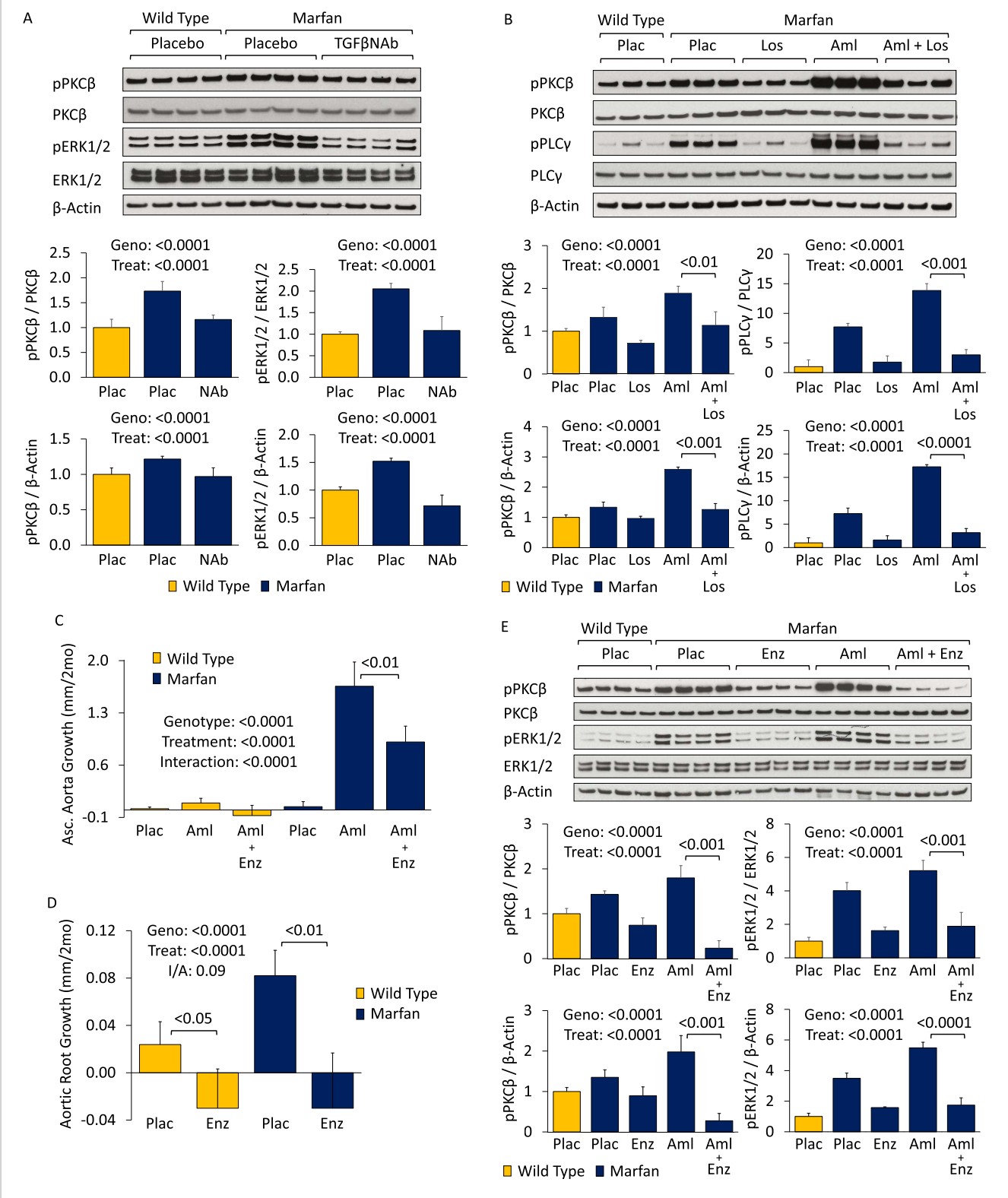

**Figure 4**. PKC activation in placebo- and CCB-treated wild-type (WT) and Marfan mice. (**A**) Western blot analysis of the aortic root and proximal ascending aorta in 4-month-old mice. Number of mice per group = 4 (2 male, 2 female). (**B**) Western blot analysis of the aortic root and proximal ascending aorta in 4-month-old mice. Number of mice per group = 3 (2 male, 1 female; or 1 male, 2 female). (**C**) Mean (±2 SEM) ascending aortic growth from 2 to 4 months of age. Number of mice per group (male/female) = WT placebo 8 (4/4), WT amlodipine 9 (4/5), WT amlodipine + enzastaurin 8 (3/5), Marfan placebo

*Figure 4. continued on next page*

Figure 4. Continued

12 (7/5), Marfan amlodipine 8 (5/3), Marfan amlodipine + enzastaurin 8 (5/3). (**D**) Mean (±2 SEM) aortic root growth from 2 to 4 months of age. Number of mice per group (male/female) = WT placebo 8 (4/4), WT enzastaurin 6 (3/3), Marfan placebo 12 (7/5), Marfan enzastaurin 8 (4/4). (**E**) Western blot analysis of the aortic root and ascending aorta in 4-month-old mice. Number of mice per group = 4 (2 male, 2 female). Plac, placebo; NAb, neutralizing antibody; Los, losartan; Aml, amlodipine; Enz, enzastaurin; Geno, genotype; Treat, treatment; I/A, interaction.

The following figure supplement is available for figure 4:

**Figure supplement 1**. PKC activation in placebo- and CCB-treated wild-type (WT) and Marfan mice.

$p < 0.0001$, *Figure 5A*). While the magnitude of rescue was greater in Marfan mice than WT littermates, this difference trended towards significance but did not quite reach it (interaction effect: $p = 0.07$).

Representative parasternal long-axis echocardiography images show the rescue achieved by hydralazine on aortic root aneurysm in Marfan mice (*Figure 5—figure supplement 1S2*). Hydralazine treatment also led to a significant rescue of aortic wall architecture, with the magnitude of effect being significantly greater in Marfan mice (*Figure 5—figure supplement 1S3*; interaction effect: $p < 0.001$, *Figure 5—figure supplement 1S4*). This correlated with a reduction of both PKCβ and ERK1/2 activation in the aortas of these animals (treatment effect: $p < 0.0001$ for both, *Figure 5B*), with the magnitude of effect again being greater in Marfan mice (interaction effect: $p < 0.001$ and $p < 0.0001$, respectively).

While PKCβ activation was greater in the aortas of Marfan mice compared to WT littermates (genotype effect: $p < 0.0001$, *Figure 5—figure supplement 1S5*), and this was exacerbated by amlodipine (treatment effect: $p < 0.0001$), RDEA119 had no significant effect on PKCβ activation (post-hoc analysis: $p = 0.44$). This suggests that PKCβ lies upstream of ERK1/2 in the pathogenic sequence of events driving aortic aneurysm progression in Marfan mice and suggests that both enzastaurin and hydralazine achieve their beneficial effect in Marfan animals via inhibition of PKCβ-mediated ERK1/2 activation (*Figure 5C*).

## Effect of CCBs in patients with Marfan syndrome

Finally, we wished to determine whether these observations made in a mouse model of Marfan syndrome hold relevance for patients. In collaboration with other members of the Genetically Triggered Thoracic Aortic Aneurysms and Cardiovascular Conditions (GenTAC) consortium (www.gentac.rti.org), we conducted a case-control study to assess the effect of CCBs on aortic dissection or aortic surgery in patients with Marfan syndrome and other forms of inherited thoracic aortic aneurysm (iTAA; *Table 1*). The other forms of iTAA primarily included Loeys-Dietz, Turner and Ehlers-Danlos syndromes, familial thoracic aortic aneurysm, and bicuspid aortic valve (BAV) with aneurysm.

Marfan patients with native aortic roots at the time of enrollment (n = 531) who received CCBs (as compared to other antihypertensive agents) had an increased odds of aortic dissection (odds ratio (OR) 12.5, $p = 0.032$). Strong trends were maintained after correction for either systolic blood pressure (OR 12.7, $p = 0.06$) or aortic root size (OR 11.2, $p = 0.08$) at enrollment. A more profound detrimental effect of CCBs on aortic dissection could be masked by the practice of performing prophylactic aortic surgery when the aorta reaches a dimension that confers risk for dissection. This is generally considered correct clinical management, since the rate of morbidity and mortality with aortic dissection is higher when surgery is performed on an emergent rather than an elective basis. In such a scenario, one might see an increased rate in the need for aortic surgery in the absence of an increased rate of aortic dissection. In keeping with this hypothesis, we found that CCB-treated Marfan patients had an increased odds of needing aortic surgery (OR 5.5, $p < 0.001$) when compared to patients on other antihypertensive agents, which remained significant when corrected for either blood pressure (OR 5.4, $p < 0.001$) or aortic size (OR 5.0, $p < 0.01$) at enrollment.

For patients with other forms of iTAA and native aortic roots at the time of enrollment (n = 1819), there was suggestion of an increased odds for aortic dissection in those who had taken CCBs, although this did not reach statistical significance (OR 4.7, $p = 0.26$). This was again most likely secondary to prophylactic surgical intervention, given that CCB-treated patients did have an increased odds of needing aortic surgery (OR 2.4, $p = 0.004$), which remained significant when corrected for either blood pressure (OR 2.2, $p = 0.016$) or aortic size (OR 2.2, $p = 0.017$) at enrollment.

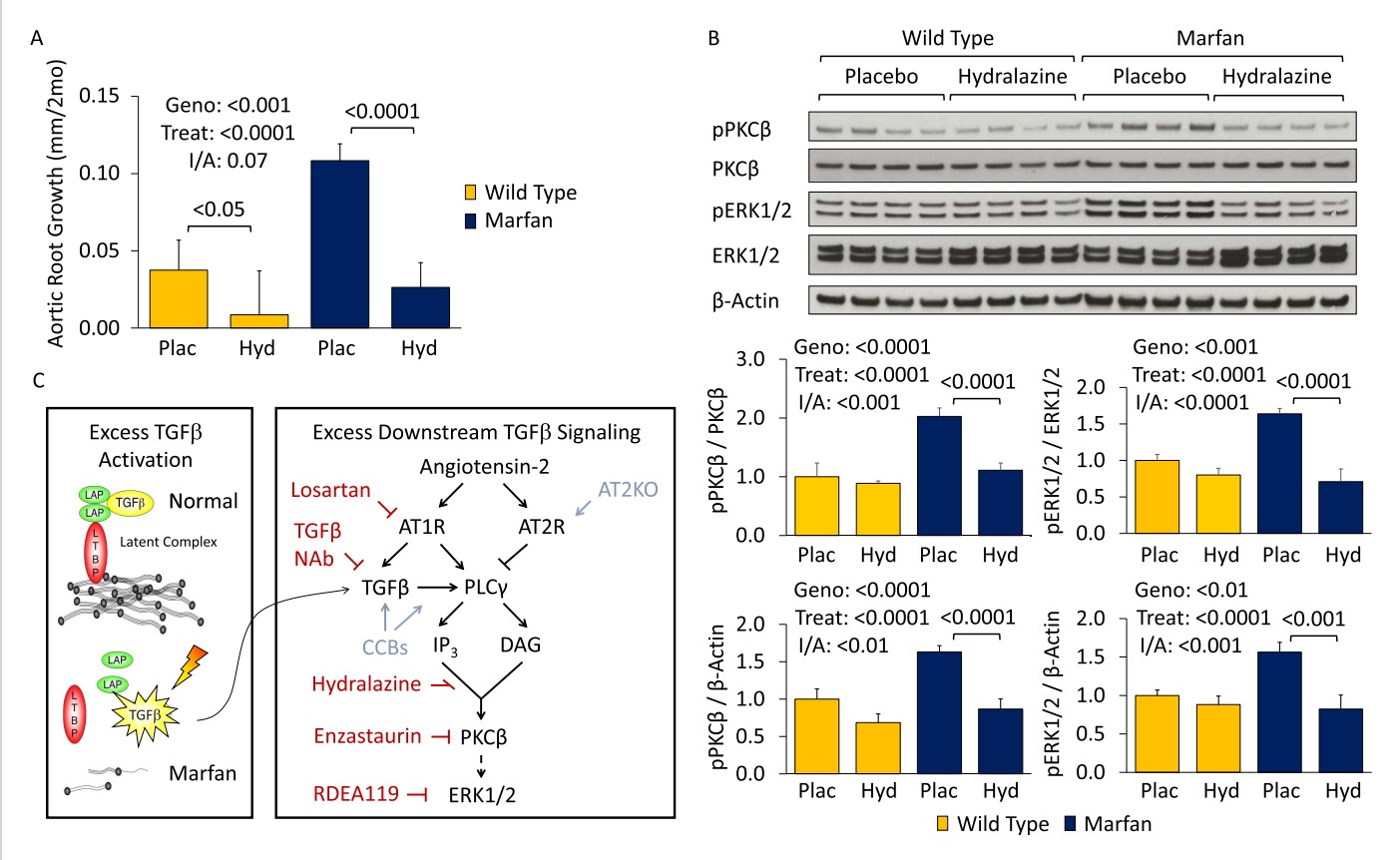

**Figure 5**. Effect of hydralazine in wild-type (WT) and Marfan mice. (**A**) Mean (±2 SEM) aortic root growth from 2 to 6 months of age. Number of mice per group (male/female) = WT placebo 9 (4/5), WT hydralazine 12 (7/5), Marfan placebo 15 (6/9), Marfan hydralazine 12 (6/6). Mean (±2 SEM) weight per group (in grams) at 6 months = 31.4 ± 2.4 g (WT placebo), 31.2 ± 3.1 g (WT hydralazine), 31.0 ± 2.7 g (Marfan placebo), 30.4 ± 3.5 g (Marfan hydralazine). (**B**) Western blot analysis of the aortic root in 6-month-old mice. Number of mice per group = 4 (2 male, 2 female). (**C**) Diagram illustrating key nodal points in Marfan mouse aortic disease pathogenesis. Drugs shown in red ameliorate aneurysm progression, while manipulations shown in blue exacerbate it. Plac, placebo; Hyd, hydralazine; Geno, genotype; Treat, treatment; I/A, interaction.

The following figure supplement is available for figure 5:

**Figure supplement 1**. Effect of hydralazine in wild-type (WT) and Marfan mice.

Given that β-blocker therapy is a first-line therapy for patients with Marfan syndrome and other related iTAA syndromes, and CCBs are typically given to patients who cannot tolerate β-blockers, one could hypothesize that the observations herein may relate more to an absence of protective β-blockade rather than use of CCBs. This interpretation is directly at odds with our mouse model data, where disease acceleration occurred due to the presence of a deleterious factor (i.e., CCB therapy), not simply due to the absence of a protective agent (i.e., β-blockade). If the latter were the case, then one would expect CCB-treated Marfan mice to have shown the same growth rate as placebo-treated Marfan animals, which was not the case. To further assess this, we controlled for β-blocker use in our GenTAC analysis (*Table 1*) and found that it did not fundamentally alter the conclusions of the study. The odds of aortic dissection and aortic surgery remained significantly increased in Marfan patients on CCBs (OR 15.9, p = 0.045; OR 5.7, p < 0.01; respectively), while the odds of aortic surgery remained significantly increased in other iTAA syndrome patients on CCBs (OR 2.0, p = 0.026).

## Discussion

Any class of drug that lowers blood pressure and/or reduces cardiac contractility may theoretically be beneficial in Marfan and related inherited TAA syndromes, since a reduction in hemodynamic stress

**Table 1.** GenTAC human data

| | Aortic dissection | | Aortic surgery | |
|---|---|---|---|---|
| | **Marfan** | **Other** | **Marfan** | **Other** |
| | **n = 531** | **n = 1819** | **n = 531** | **n = 1819** |
| Odds in CCB | 5.1% | 0.57% | 28.1% | 10.70% |
| Odds in non-CCB | 0.41% | 0.12% | 5.1% | 4.4% |
| Odds ratio | 12.5 | 4.7 | 5.5 | 2.4 |
| p-value | 0.032 | NS | <0.001 | <0.01 |
| Odds ratio (BP)* | 12.7 | 5.6 | 5.4 | 2.2 |
| p-value | 0.06 | NS | <0.001 | 0.016 |
| Odds ratio (Aortic Size)* | 11.2 | 4.1 | 5.0 | 2.2 |
| p-value | 0.08 | NS | <0.01 | 0.017 |
| Odds ratio (β-blocker)* | 15.9 | 3.7 | 5.7 | 2.0 |
| p-value | 0.045 | NS | <0.01 | 0.026 |

Odds of 'aortic dissection' and 'aortic surgery' in patients with Marfan syndrome ('Marfan') and other forms of inherited thoracic aortic aneurysm ('Other'). Odds (written as %) = number of people who incurred an event (i.e., dissection or surgery) divided by the number who did not. Odds of aortic dissection or surgery were calculated separately for patients who had used CCBs ('Odds in CCB') compared to those who had not (i.e., 'Odds in non-CCB'). Odds Ratio = Odds of aortic dissection or surgery in patients who had taken CCBs divided by the odds in patients who had not taken CCBs.

*The Odds Ratio was then adjusted for blood pressure ('BP'), aortic size ('Aortic Size'), and β-blocker use ('β-blocker') at enrollment, with corresponding p-values.

placed upon an inherently weakened aortic wall should hypothetically limit aortic expansion and delay or prevent aortic dissection. Prior studies in mouse models of Marfan syndrome and related disorders have shown that β-blockers (e.g., propranolol) (*Habashi et al., 2006*; *Gallo et al., 2014*) or angiotensin converting enzyme inhibitors (ACEi, e.g., enalapril) (*Habashi et al., 2011*) achieve a relatively modest inhibition of aortic aneurysm growth and fail to preserve aortic wall architecture when compared to the ARB losartan, despite an equivalent reduction in blood pressure. Notably, in Marfan mice the extent of relative protection of these agents correlates closely with suppression of TGFβ-dependent canonical (Smad2/3) or noncanonical (ERK1/2) signaling (*Holm et al., 2011*; *Cook et al., 2015*). While these different classes of antihypertensive agent show varying degrees of protective effect in Marfan mice, they all confer at least some degree of protection against aneurysm progression.

To our knowledge, this is the first time that a class of clinically available blood pressure lowering agents has been shown to exacerbate aortic aneurysm and cause dissection in a mouse model of Marfan syndrome. This is despite the fact that amlodipine lowered blood pressure equally as much as losartan, an effect which should provide some degree of protection against aneurysm progression. This deleterious effect on aneurysm growth was not limited to the dihydropyridine amlodipine, but also extended to another class of CCB, namely the phenylalkylamine verapamil. Given that a significant number of patients with Marfan syndrome and related conditions are prescribed CCBs, a previously unidentified deleterious role for these drugs has important clinical ramifications.

It is interesting to note that amlodipine and verapamil showed the same trend, namely a small deleterious effect in WT mice and a much greater accentuation of growth in Marfan mice, with the effect being greater in the ascending aorta than the aortic root. In terms of absolute growth, amlodipine had a significantly greater effect than verapamil, even at its lower dose. While the two drugs both target L-type calcium channels on the cell membrane, verapamil is generally considered to have greater selectivity for cardiac tissue, while amlodipine is considered to have stronger tropism for aortic VSMCs. It may be for this reason that amlodipine has a relatively greater detrimental effect on the aorta compared to verapamil.

We have further elucidated that PKCβ-mediated ERK1/2 activation contributes to the deleterious effect of CCBs in Marfan mice, while inhibition of this TGFβ- and AT1R-dependent pathway using either a PKC inhibitor (enzastaurin) or the clinically available antihypertensive agent hydralazine is able

to prevent aortic aneurysm progression in Marfan mice, in association with blunted PKCβ and ERK1/2 activation. While ARBs such as losartan and ACE inhibitors such as enalapril are teratogenic and hence not appropriate for use in pregnancy, hydralazine is well tolerated, making it an appealing alternative therapeutic strategy for Marfan syndrome, particularly in pregnant women.

It is also notable that while CCBs accelerate growth of the aortic root in Marfan mice, they have an even more pronounced effect on the more distal ascending aorta, an aortic segment that is less frequently affected in Marfan syndrome. The molecular basis of regional predisposition for aortic aneurysm is not yet fully understood but may relate to the distinct developmental origins of the two aortic segments. While vascular smooth muscle cells in the aortic root primarily derive from the second heart field (SHF), those in the more distal ascending aorta derive from the cardiac neural crest (CNC). Prior work has shown that L-type calcium channel function is critical for correct CNC migration, differentiation, and morphogenic patterning, as well as maintenance of an appropriate post-developmental differentiated state (*Moran, 1991*). Furthermore, mutations in L-type calcium channels result in cellular hypertrophy and hyperplasia in neural-crest-derived tissues (*Ramachandran et al., 2013*). This may explain why CCBs appear to have a greater deleterious effect on the ascending aorta compared to the aortic root.

Interestingly, the ascending aorta is characteristically involved in patients with BAV with aneurysm. Despite being the most common developmental cardiovascular abnormality in humans, the genetic cause(s) and molecular mechanisms underlying BAV and the associated aortopathy are not well understood. This work provides rationale and incentive to elucidate whether altered calcium, PKC and/or ERK1/2 signaling may play a role in this condition.

There is only one prior study assessing the effect of CCBs on aortic growth in Marfan patients (*Rossi-Foulkes et al., 1999*). The number of CCB-treated patients was small (n = 6), and the study combined data on patients treated with either CCBs or β-blockers. Given that β-blockers are known to confer protection against aortic growth in Marfan patients (*Salim et al., 1994*; *Shores et al., 1994*; *Silverman et al., 1995*), and nearly 80% of patients in the study were on β-blockers but not CCBs, a detrimental effect of CCBs could easily have been masked. Interestingly in the study, the complication rate was more than twice as high in CCB-treated patients as compared to those on β-blockers (33% vs 15%), although numbers were too small to draw firm conclusions.

While randomized double-blinded prospective clinical trials are the preferred approach to analyze the therapeutic efficacy of a drug, this is not really feasible for agents that are found to be severely detrimental in pre-clinical models, since ethical approval for a prospective human trial is unlikely to be granted. Retrospective analyses of large data sets are a more realistic solution in such scenarios. An inherent limitation to any analysis of rare Mendelian disorders is the number of patients available. This is even more challenging when stratifying patients based on drugs that have been prescribed. The GenTAC consortium was established to try to overcome some of these challenges. However, it needs to be recognized that the human data contained herein is still limited by relatively small sample size (more so for aortic dissection than aortic surgery) and an inability to control for the dose or number of medications that patients were receiving. We were able to control for aortic size at the time of enrollment, which is an indicator of baseline aortic disease severity and a predictor of future risk of aortic dissection and/or need for aortic surgery. We were also able to control for blood pressure, which is a known risk factor for aneurysm progression. Hence while the conclusions that can be drawn are not definitive, the data suggest that CCBs should be used with caution in Marfan patients. They also suggest that CCBs may be deleterious in other Marfan-related conditions, although a larger sample size will be needed to assess risk after stratification by individual disorders.

## Materials and methods

### Mice

All mice were cared for under strict compliance with the Animal Care and Use Committee of the Johns Hopkins University School of Medicine. The *Fbn1*^C1039G/+ line was maintained on a pure C57BL/6J background (backcrossed for >12 generations), allowing for valid comparisons. Mice were sacrificed with an inhalation overdose of halothane (Sigma–Aldrich, St. Louis, MI, United States). Mice underwent immediate laparotomy, descending abdominal aortic transection, and phosphate-buffered saline (PBS) (pH 7.4) was infused through the right and left ventricles to flush out the blood. Mouse

aortic root and ascending aortas (aortic root to origin of right brachiocephalic trunk) were harvested, snap-frozen in liquid nitrogen and stored at minus 80° centigrade until processed. Protein was extracted using the reagents and protocol from a Total Protein Extraction Kit containing protease inhibitor and Protein Phosphatase Inhibitor Cocktail (Millipore, MA, United States). Aortas were homogenized using a pellet pestle motor (Kimble-Kontes, NJ, United States) as per the extraction kit protocol. Samples were then stored once more at minus 80° centigrade until Western blot analysis was performed. Mice that were analyzed for aortic histology had latex infused into the left ventricle at a pressure between 70 and 80 mmHg, as confirmed using a handheld digital manometer (Fisher Scientific, Pittsburgh, PA, United States). Mice were then fixed for 24 hr in 10% buffered formalin, after which time the heart and aorta were removed and stored in 70% ethanol until histological analysis was performed.

## Drug treatment

Mice were started on medication at 8 weeks of age. Losartan was dissolved in drinking water and filtered to reach a concentration of 0.6 g/l, giving an estimated daily dose of 60 mg/kg/day (based on a 30 g mouse drinking 3 mls per day). Amlodipine was dissolved in drinking water and filtered to reach a final concentration of 0.12 g/l, giving an estimated daily dose of 12 mg/kg/day. Verapamil was dissolved in drinking water and filtered to reach a concentration of 1.44 g/l, giving an estimated daily dose of 144 mg/kg/day. Hydralazine was dissolved in drinking water and filtered to reach a concentration of 0.16 g/l, giving an estimated daily dose of 16 mg/kg/day. Placebo-treated animals received drinking water. RDEA119 (25 mg/kg) and enzastaurin (15 mg/kg) were reconstituted in 10% 2-hydroxypropyl-beta-cyclodextrin (Sigma–Aldrich) dissolved in PBS and administered twice daily by oral gavage. Treatment for both was initiated at 8 weeks of age and continued for 8 weeks. 10% 2-hydroxypropyl-beta-cyclodextrin dissolved in PBS was used as the placebo control. Mouse monoclonal TGFβ NAb (1d11; R&D Systems, Minneapolis, MN, United States) was reconstituted in PBS and administered via intraperitoneal injection three times a week at a dose of 5 mg/kg. Treatment was initiated at 1 month of age and continued for 2 months. IgG (Zymed Laboratories Inc, San Francisco, CA, United States) was reconstituted in PBS, and administered at a dose of 10 mg/kg as a control.

## Echocardiography

Nair hair removal cream was used on all mice the day prior to echocardiograms. All echocardiograms were performed on awake, unsedated mice using the Visualsonics Vevo 2100 and a 30 MHz transducer. Mice were imaged at baseline and every 2 months after treatment until the time of sacrifice. The aorta was imaged using a parasternal long axis view. Three separate measurements of the maximal internal dimension at the sinus of Valsalva and proximal ascending aorta were made from distinct captured images and averaged. All imaging and measurements were performed blinded to both genotype and treatment arm.

## Blood pressure and heart rate

Blood pressures and heart rates were measured by tail cuff plethysmography the week prior to completion of a study. Mice were habituated to the system for 5 days and then on the final day 3–5 measurements were obtained and averaged. 8 mice for each treatment group were analyzed.

## Histological and morphometric analysis

Latex-infused ascending aortas were transected just above the level of the aortic valve, and 3-mm transverse sections were mounted in 4% bacto-agar prior to paraffin fixation. Five micrometer aortic sections underwent Verhoeff-van Giesen (VVG) and Masson's Trichrome staining and were imaged at 40× magnification, using a Nikon Eclipse E400 microscope. Wall architecture of 4 representative sections for each mouse was assessed by 4 blinded observers and graded on an scale of 1 (indicating normal histology) to 5 (indicating diffuse elastic fiber fragmentation and histological damage), and the results were averaged.

## Western blot analysis and antibodies

Aortic tissue homogenates were dissolved in sample buffer, run on a NuPAGE Novex 4–12% Bis-Tris Gel (Invitrogen, CA, United States), and transferred to nitrocellulose membranes using the iBlot transfer system (Invitrogen). Membranes were washed in PBS and blocked for 1 hr at room

temperature with 5% instant non-fat dry milk dissolved in PBS containing 1% Tween-20 (Sigma, MO, United States) (PBS-T). Equal protein loading of samples was determined by a protein assay (Bio-Rad, CA, United States) and confirmed by probing with antibodies against β-Actin (Sigma). Membranes were probed overnight at 4° centigrade with primary antibodies against pSmad3 (1880-1; Millipore), Smad3 (#9513; Cell Signaling), pERK1/2 (#4370; Cell Signaling), ERK1/2 (#4695; Cell Signaling), pPKCβ (#75837; Abcam, United Kingdom), PKCβ (#32026; Abcam), pPLCγ (#2821; Cell Signaling), and PLCγ (#2822; Cell Signaling), dissolved in PBS-T containing 5% milk. Blots were then washed in PBS-T and probed with HRP-conjugated anti-rabbit secondary antibody (GE Healthcare, United Kingdom) dissolved in PBS-T containing 5% milk at room temperature. Blots were then washed in PBS-T, developed using SuperSignalWest HRP substrate (Pierce Scientific, IL, United States), exposed to BioMax Scientific Imaging Film (Sigma), and quantified using ImageJ analysis software (NIH, MD, United States).

## Human aortic dissection and surgery analysis

This was performed using clinical data from the GenTAC registry. In short, patients were recruited from 5 regional clinical centers that cover a wide geographic catchment area within the United States, including Johns Hopkins University, Oregon Health & Science University, University of Pennsylvania, University of Texas Health Science Center at Houston/Baylor College of Medicine, and Weill Cornell Medical College. Each site obtained approval to conduct the study from their respective institutional review boards, and informed consent was obtained at each site. Longitudinal observational data were collected on adults and children diagnosed with one of 12 thoracic aortic aneurysm-related conditions, primarily including Marfan, Loeys-Dietz, Ehlers-Danlos and Turner syndromes, BAV, and familial thoracic aortic aneurysm. Designated registry investigators at each enrolling site confirmed diagnostic classifications of genetically associated aortic conditions. Demographic indices were abstracted from medical records at the time of registry enrollment by site-specific research coordinators, and all samples were de-identified to preserve patient confidentiality.

The current analysis was limited to aortic dissection or aortic surgery that occurred in the aortic root, ascending aorta and/or aortic arch, and excluded patients who had had aortic dissection or surgery in these regions prior to enrollment into the study. CCB use was considered positive if patients had taken them prior to, or were taking them at the time of, enrollment into the study. Aortic dissection or surgery was assessed prospectively in the follow-up period after enrollment. Mean follow-up was 50.8 ± 1.6 months for the Marfan patients and 43.4 ± 0.8 months for all TAA syndrome patients. Statistical analyses were performed using SAS Version 9.3 (SAS Institute, Cary, NC, United States). Exact ORs and two-sided p-values were calculated using PROC LOGISTIC. Models were adjusted for potential cofounders identified a priori (i.e., systolic blood pressure measurement and aortic root size measurement at the time of enrollment). Interaction terms were determined not to be statistically significant ($p > 0.05$), so only main effects were included in the final models.

## Statistical analysis

All quantitative data are shown as bar graphs produced using Excel (Microsoft, Redmond, WA, United States). Mean ±2 standard errors of the mean (SEM) are displayed. Statistical analysis was performed using two-way ANOVA for all continuous data with three or more groups and two potentially interacting terms (e.g., echocardiography, Western blot); Kruskal–Wallis ANOVA was used to analyze categorical data with three or more groups and two potentially interacting terms (e.g., aortic architecture score); one-way ANOVA was used to analyze continuous data with three or more groups but no interaction (e.g., blood pressure); two-tailed $t$ tests were used to analyze data comparing two groups, or to make selective planned comparisons between individual groups within a larger study. Significance values for the effects of genotype, treatment, and/or any interaction between two variables have been included in each figure, where appropriate. If only placebo treatment for WT mice was included in an analysis, no interaction between drug treatment and genotype could be assessed, so it is not included in the figure. A p value $< 0.05$ was considered statistically significant in all analyses.

## Acknowledgements

This work was supported by NIH (HCD, DPJ); Howard Hughes Medical Institute (HCD, AJD); National Marfan Foundation (HCD, JPH, JJD); Cellular and Molecular Medicine Training Program, Johns Hopkins School of Medicine (JJD, NCW); Smilow Center for Marfan Syndrome Research and MIBAVA Leducq Consortium (HCD).

## Additional information

### Group author details

**GenTAC Registry Consortium**

Carrie Farrar: Oregon Health and Science University, Portland, Oregon; Williams Ravekes: Johns Hopkins University, Baltimore, United States; Harry C Dietz: Johns Hopkins University, Baltimore, United States; Kira Lurman: Johns Hopkins University, Baltimore, United States; Kathryn W Holmes: Johns Hopkins University, Baltimore, United States; Jennifer Habashi: Johns Hopkins University, Baltimore, United States; Dianna M Milewicz: University of Texas, Houston, United States; Siddharth K Prakash: University of Texas, Houston, United States; Meghan Terry: University of Texas, Houston, United States; Scott A LeMaire: Baylor College of Medicine, Houston, United States; Shaine A Morris: Baylor College of Medicine, Houston, United States; Irina Volguina: Baylor College of Medicine, Houston, United States; Cheryl L Maslen: Oregon Health and Science University, Portland, Oregon; Howard K Song: Oregon Health and Science University, Portland, Oregon; G Michael Silberbach: Oregon Health and Science University, Portland, Oregon; Reed E Pyeritz: University of Pennsylvania, Philadelphia, United States; Joseph E Bavaria: University of Pennsylvania, Philadelphia, United States; Karianna Milewski: University of Pennsylvania, Philadelphia, United States; Amber Parker: University of Pennsylvania, Philadelphia, United States; Richard B Devereux: Weill Medical College, Cornell University, New York, United States; Jonathan W Weinsaft: Weill Medical College, Cornell University, New York, United States; Mary J Roman: Weill Medical College, Cornell University, New York, United States; Tanya LaTortue: Weill Medical College, Cornell University, New York, United States; Ralph Shohet: The Queen's Medical Center, Honolulu, United States; Fionna Kennedy: The Queen's Medical Center, Honolulu, United States; Nazli McDonnell: National Institute on Aging, United States; Ben Griswold: National Institute on Aging, United States; Federico M Asch: MedStar Health Research Institute, Hyattsville, United States; Neil J Weissman: MedStar Health Research Institute, Hyattsville, United States; Kim A Eagle: University of Michigan, Detroit, United States; H Eser Tolunay: National Heart, Lung, and Blood Institute, United States; Patrice Desvigne-Nickens: National Heart, Lung, and Blood Institute, United States; Mario P Stylianou: National Heart, Lung, and Blood Institute, United States; Megan Mitchell: National Heart, Lung, and Blood Institute, United States; Hung Tseng: National Institute of Arthritis, Musculoskeletal and Skin Diseases, United States; Barbara L Kroner: RTI International, Research Triangle Park, United States; Tabitha Hendershot: RTI International, Research Triangle Park, United States; Ryan Whitworth: RTI International, Research Triangle Park, United States; Danny Ringer: RTI International, Research Triangle Park, United States; Liliana Preiss: RTI International, Research Triangle Park, United States; Meg Cunningham: RTI International, Research Triangle Park, United States; Natalia Bradley: RTI International, Research Triangle Park, United States

### Group author details

**MIBAVA Leducq Consortium**

Harry C Dietz: Johns Hopkins University, Baltimore, United States; Andrew S McCallion: Johns Hopkins University, Baltimore, United States; Bart Loeys: University of Antwerp, Antwerp, Belgium; Lut Van Laer: University of Antwerp, Antwerp, Belgium; Per Eriksson: Karolinska Institute, Stockholm, Sweden; Anders Franco-Cereceda: Karolinska Institute, Stockholm, Sweden; Luc Mertens: Sick Kids University Hospital, Toronto, Canada; Seema Mittal: Sick Kids University Hospital, Toronto, Canada; Salah A Mohamed: Lübeck University Hospital, Lübeck, Germany; Gregor Andelfinger: Sainte-Justine Hospital, Montréal, Canada

### Competing interests

HCD: Reviewing editor, *eLife*. The other authors declare that no competing interests exist.

## Funding

| Funder | Grant reference | Author |
|---|---|---|
| National Institutes of Health | AR041135 | Jennifer P Habashi, Harry C Dietz |
| Howard Hughes Medical Institute | | Alexander J Doyle, Harry C Dietz |
| National Marfan Foundation | Bloomberg Fund | Jefferson J Doyle, Jennifer P Habashi, Harry C Dietz |
| National Marfan Foundation | Smilow Center for Marfan Syndrome Research | Harry C Dietz |
| Fondation Leducq | MIBAVA Leducq Consortium | MIBAVA Members |
| National Heart, Lung, and Blood Institute | GenTAC Consortium | GenTAC Members |
| National Institute of Health | AR049698 | Jennifer P Habashi, Harry C Dietz |

The funders had no role in study design, data collection and interpretation, or the decision to submit the work for publication.

## Author contributions

JJD, Final approval of the version to be published, Conception and design, Acquisition of data, Analysis and interpretation of data, Drafting or revising the article; AJD, MEL, Final approval of the version to be published, Acquisition of data, Analysis and interpretation of data, Drafting or revising the article; NKW, JPH, DB, LM, NH, SB, OS, BR, HE, DH, Final approval of the version to be published, Acquisition of data, Drafting or revising the article; REW, FS, Final approval of the version to be published, Analysis and interpretation of data, Drafting or revising the article; CJT, DPJ, Final approval of the version to be published, Conception and design, Drafting or revising the article, Contributed unpublished essential data or reagents; MJC, JEVE, Final approval of the version to be published, Drafting or revising the article, Contributed unpublished essential data or reagents; HCD, Conception and design, Analysis and interpretation of data, Drafting or revising the article

## Ethics

Human subjects: For the GenTAC registry, patients were recruited from 5 regional clinical centers within the United States, including Johns Hopkins University, Oregon Health & Science University, University of Pennsylvania, University of Texas Health Science Center at Houston/Baylor College of Medicine, and Weill Cornell Medical College. Each site obtained approval to conduct the study from their respective institutional review boards, and informed consent was obtained at each site. Demographic indices were abstracted from medical records at the time of registry enrollment by site-specific research coordinators, and all samples were de-identified to preserve patient confidentiality. RTI was the data coordinating center for the analysis. The GenTAC protocol number at RTI was 11832.

Animal experimentation: This study was performed in strict accordance with the recommendations in the Guide for the Care and Use of Laboratory Animals of the National Institutes of Health. All mice were cared for under strict compliance with the Animal Care and Use Committee of the Johns Hopkins University School of Medicine. The research protocols under which this work was conducted were: MO09M112, MO12M124 and MO15M88.

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
