## [Decision Letter]

Thank you for submitting your work entitled “A Deleterious Gene-by-Environment Interaction Imposed by Calcium Channel Blockers in Marfan Syndrome” for peer review at *eLife*. Your submission has been favorably evaluated by Sean Morrison (Senior Editor) and three reviewers, one of whom is a member of our Board of Reviewing Editors.

The reviewers have discussed the reviews with one another and the Reviewing Editor has drafted this decision to help you prepare a revised submission.

This manuscript describes intriguing and provocative and important results, both in a mouse model and with retrospective outcome studies in humans, that calcium channel blocker therapy adversely affects the natural course of disease in Marfan's syndrome.

All three reviewers considered your paper to be an impressive and important piece of work, one that is potentially suitable for publication in *eLife*. However, each of the three reviewers had significant criticisms that must be addressed before your manuscript can be considered acceptable for publication.

The comments by Reviewer #1 need to be addressed in full. Reviewer #1 was mainly concerned about missing pieces of data in your manuscript.

Reviewer #2 had similar concerns about the completeness of the data. For example, you need to include latex-injected images of hydralzine-treated mice. Also, you should show ascending aorta measurements in Figure 4 and Figure 5. Reviewer #2 also suggested that you discuss the mechanism of action of hydralazine.

Reviewer #2 also was intrigued by the finding that amlodipine and verapamil affected the ascending aorta more than the aortic root. You need to discuss potential mechanisms and provide experimental data, if possible.

Also, both Reviewers #2 and #3 questioned the validity of your schema in Figure 5. AT1R was recently been shown to be activated in a ligand-independent manner, and the authors have not formally addressed angiotensin-2 dependency in the AT1R activation. These issues should be addressed with experiments (or if this is not feasible then at least discussed).

Reviewers #2 and #3 had major concerns about the western blots, and particularly the normalization of western blot data. For western blot analyses of phosphorylated signaling molecules, you should normalize the signals to their nonphosphorylated forms or to the total protein, not to β-actin.

Reviewers #2 and #3 were also concerned about the statistical analyses, including use of *t*-tests for experiments with multiple groups or categorical data and failure to use 2-way ANOVA or another approach to evaluate the significance of any interactions between drug and genotype.

Reviewer #3 also challenged your interpretation of the observational data in humans. β-blockers are a first-line therapy and CCBs might normally be given to patients who could not tolerate β-blockers. For this reason, you must find a way to include the use of β-blockers in your analyses.

Reviewer #3 had other important criticisms. Your figures should ideally use consistent units and axes so that they can be compared easily. Also, you need to defend the dosages of medicines that were given to mice and discuss whether the high doses in mice are relevant to humans. Can you discuss plasma levels in mice and humans?

You should provide a point-by-point response to all of the reviewers' criticisms.

Reviewer #1:

This is a very impressive paper. It contains a very large amount of data. The data are convincing. The findings are clinically important. The pharmacologic observations add significantly to our understanding of signaling pathways in Marfan's syndrome. The basic science observations in mice were confirmed with a retrospective analysis of outcomes in human patients.

All of my comments are relatively minor (but still important):

1) The table is easy to understand when reading the text, but hard to follow on its own.

2) I think that you should choose a more traditional format for the figure legends. Leave the “results” out of the figure legends.

3) Blood pressure should be reported for the hydrazine group.

4) Report body weights for mice during the study. Any data on heart rate?

5) Ages and sexes of mice for each group should appear in the legend.

6) Numbers of mice for each group in each experiment should be reported in each figure legend.

7) Exact P value for 12.5 odds ratio in table should be reported (not just <0.05).

Reviewer #2:

The manuscript by Doyle at el. addresses a clinically important problem associated with drug treatment for Marfan patients and related disorders. The authors presented evidence that administration of calcium channel blockers (CCB) enhanced growth of ascending aortas and induced dissection in Marfan model mouse (*Fbn1*^*C1039G/+*^) by increasing PKCβ and PLCγ in TGFβ- and AT1R-dependent manner. The exacerbated phenotype was rescued by administration of PKC inhibitor or hydralazine, which was proposed to inhibit PKC-dependent ERK phosphorylation. Lastly, the authors used GenTAC consortium database to show that patients on CCB had a stronger trend of aortic dissection even after correction for systolic blood pressure or aortic size. The information obtained from this study is critical for consideration of drug regimen for Marfan patients. It also provides an opportunity for hydralazine protocol. Overall, the message is clearly delivered, however, additional information will strengthen the manuscript.

1) Both Amlodipine and Verapamil affect ascending aortas more than aortic root regions. Can the authors offer any explanation? Collagen is dramatically increased after CCB treatment yet the aortic wall was more prone to dissection. Do the authors have any ideas as to which collagens are induced and whether they are functional in CCB-treated aortas? Are fibroblasts activated in CCB-treated animals in a manner similar to gingival overgrowth caused by CCB?

2) What is hydralazine's mechanism of action on VSMC?

3) There are no latex-injected images of hydralazine-treated aortas. Please provide.

4) Since ascending aorta showed more severe phenotype, the authors should show ascending aorta measurements in Figure 4 and Figure 5.

5) In contrast to schema in Figure 5, AT1R was shown to be activated in a ligand-independent manner (Cook JR et al., JCI, 2014). The authors have not formally addressed angiotensin-2 dependency in the AT1R-activation.

Reviewer #2 (Additional data files and statistical comments):

Appropriate statistical analyses (other than *t*-test) should be performed for the comparison of samples more than 3.

Reviewer #3:

This is a provocative and potentially important study that seems to show both detrimental and beneficial effects of clinically available antihypertensive drugs in a mouse model of MFS. Effects of these drugs are documented macroscopically (aortic diameter), microscopically (histologic sections), and biochemically (western blotting). These data are supplemented by experiments that use pharmacologic antagonists to elucidate pathways through which these antihypertensives might be acting. Finally, a retrospective case-control study attempts to extend one of the major findings (detrimental effect of amlodipine) to humans with MFS or other forms of inherited TAA. The data here potentially expand our understanding of the pathogenesis of MFS-associated aortopathy and may also help guide medical therapy for MFS.

1) Doses of amlodipine, verapamil, and hydralazine (in mg/kg/d) are much higher than those used in humans: ∼100-fold (the lower amlodipine dose is still 25-fold), ∼40-fold, and ∼5-fold, respectively. Authors must comment on the relevance of these murine doses to human therapeutics. Off-target effects seem likely. Relevant to this, Figure 1 and Figure 1—figure supplement 1 show that this dose of amlodipine increases aortic growth in wild-type mice by several-fold. Similarly verapamil increases aortic growth in wild-type mice by ∼10-fold. This is highly unlikely to apply to humans taking amlodipine and verapamil; why should the results in MFS mice be any more translatable to humans taking these drugs?

2) The figures should use consistent units and axes so results can be easily compared. Aortic growth should be in mm/2mo only. The scale for all aortic root graphs should be identical in all figures as should the scale for all ascending aorta graphs.

3) In the subsection “Effect of Non-Dihydropyridine CCBs: Verapamil,” ascending aortic growth in verapamil-treated MFS mice did not “closely parallel” that in amlodipine-treated mice. It was much less (∼ 75% less). Maybe this is because the verapamil dose was only 40 times the typical human dose but the amlodipine dose was 100 times the typical human dose.

4) All of the western analyses for phosphorylated signaling molecules must be normalized to their unphosphorylated forms, not to β-actin. Also, in Figure 3, pSmad3 is the upper not the lower band.

5) Ascending aortic growth is minimally variable within an experiment, but highly variable between experiments (compare Figures 1 and 3). I have the same concern for aortic root growth (compare Figures 1 and 4). Is there a good explanation?

6) Figure 5 seems at odds with earlier work from this group in which TGF-β is identified as the primary initiator of aortopathy in MFS mice (due to excessive activation associated with mutant fibrillin). Here, angiotensin-2 is portrayed as the primary initiator, with TGF-b activation downstream. Also, the figure suggests that aortopathy could occur independently of TGF-β. Please explain.

7) The GenTac analyses seem to have a fundamental flaw. Because β-blockers are first-line therapy and CCBs would typically be given to patients who could not tolerate β-blockers, the observations here may be entirely due to lack of β blockade rather than use of CCBs. To avoid this, the analyses here must either be controlled for β-blocker use or, alternatively, a propensity-matching approach must be used.

Reviewer #3 (Additional data files and statistical comments):

1) Relevant to concern #1, the authors must objectively test whether the agents produce different results in wild-type versus MFS mice. This would be done by 2-way ANOVA, specifically the interaction term that tests whether (for example in Figure 1) the effect of amlodipine on aortic growth depends on genotype. This is particularly important because the title reports a “gene by environment interaction” whereas no statistical support for a significant interaction is present in this manuscript.

2) The statistics section reports only use of the 2-tailed *t*-test. The aortic architecture data, however, is obtained with a categorical not a continuous scale and should be analyzed with nonparametric tests (rank-sum and Kruskal-Wallis ANOVA). In addition, *t*-tests are used for multi-group experiments. This is not proper.

3) Figure legends should all report the number of animals per group.

4) With regards Figure 3—figure supplement 1, did RDEA have any effect on pSmad3?

---

## [Author Response]

Reviewer #1:

1) The table is easy to understand when reading the text, but hard to follow on its own.

The table and legend have been amended to make it as simple as possible to understand.

2) I think that you should choose a more traditional format for the figure legends. Leave the “results” out of the figure legends.

These have been removed from the legends as requested.

3) Blood pressure should be reported for the hydrazine group.

Blood pressure and heart rate data for hydralazine have now been included in the figures (Figure 5).

4) Report body weights for mice during the study. Any data on heart rate?

Body weight data has now been included in the figure legends for the main studies (amlodipine, verapamil, hydralazine). Heart rate data has now been included alongside the blood pressure data (Figure 1—figure supplement 1 and Figure 5).

5) Ages and sexes of mice for each group should appear in the legend.

The ages and sexes of the mice have now been included in the figure legends for each trial.

6) Numbers of mice for each group in each experiment should be reported in each figure legend.

The number of mice in each group of a trial has now been included in the figure legends.

7) Exact P value for 12.5 odds ratio in table should be reported (not just <0.05).

This has been updated in both the text and Table 1 to reflect the precise p-value (0.032).

Reviewer #2:

1) Both Amlodipine and Verapamil affect ascending aortas more than aortic root regions. Can the authors offer any explanation? Collagen is dramatically increased after CCB treatment yet the aortic wall was more prone to dissection. Do the authors have any ideas as to which collagens are induced and whether they are functional in CCB-treated aortas? Are fibroblasts activated in CCB-treated animals in a manner similar to gingival overgrowth caused by CCB?

The molecular basis of regional predisposition for aortic aneurysm is a fascinating question that has yet to be resolved. The aortic root is developmentally distinct from the ascending aorta. While vascular smooth muscles cells (VSMCs) in the aortic root primarily derive from the second heart field (SHF), those in the more distal ascending aorta derive from the cardiac neural crest (CNC). Prior work has shown that neural crest cells express L-type calcium channels during migration and differentiation, while blockage of these channels by CCBs, both in vivo and in vitro, leads to a dramatic and reversible inhibition of neural crest migration (18). Furthermore, CCB treatment of whole embryos, even after neural crest cells have already migrated and differentiated, still causes a significant change in individual cell shape and morphogenic patterning, suggesting that maintenance of a differentiated state in neural crest cells may be regulated at least in part by -L-type calcium channels. Other work has shown that mutations in L-type calcium channels result in cellular hypertrophy and hyperplasia in neural- crest derived tissues (24). Our current hypothesis is that SHF- and CNC-derived VSMCs show variable responses to the underlying disease process and to CCB treatment in this context. We are currently attempting to test this hypothesis, but are hampered by the lack of adult markers of VSMC lineage – necessitating a complex breeding strategy that introduces lineage-specific marker alleles. Increased vessel wall collagen is a common finding in inherited aneurysm conditions that have been linked to TGFβ. Our current belief is that collagen deposition is a marker of other pathogenic events, but does not contribute to disease progression. Given our rudimentary understanding of these issues, any proposed mechanistic connection to the pathogenesis of gingival hyperplasia in CCB-treated patients would be overly speculative at this time, but worthy of future study.

2) What is hydralazine's mechanism of action on VSMC?

Hydralazine has been proposed to have several mechanisms of action. Prior work has shown that it can inhibit IP3-mediated calcium release from the sarcoplasmic reticulum (SR)(Gurney and Allam, 1995; Ellershaw and Gurney, 2011). There is also evidence that hydralazine can inhibit PKC activation and its downstream phosphorylation of ERK in T cells (8). We could find no prior studies linking hydralazine to inhibition of PKC-mediated ERK activation in VSMCs, suggesting that this may be the first report of it.

3) There are no latex-injected images of hydralazine-treated aortas. Please provide.

Echocardiography is preferable to image the aortic root, which is embedded in the outflow tract of the left ventricle, while latex injection is a better modality to highlight the more distal ascending aorta. Given that hydralazine rescued aortic root growth but did not cause ascending aortic aneurysm, we did not perform latex injections on this cohort. We have now included representative primary echocardiography images of the aortic root in placebo- and hydralazine-treated wild type and Marfan mice from this experiment (Figure 5—figure supplement 1).

*4) Since ascending aorta showed more severe phenotype, the authors should show ascending aorta measurements in*
Figure 4
*and*
Figure 5*.*

Figures 4 and 5 involve Marfan mice in the absence of CCB treatment. Given that these mice show aortic root but not ascending aortic aneurysm (genotype effect: p=0.72 and p=0.99), there was limited opportunity for a therapeutic effect of either enzastaurin or hydralazine on the ascending aorta. Since this was deemed not to be a particularly informative result, we left these data out of the manuscript. We have provided Figures 6 and 7 for inspection by the reviewer, and would be happy to incorporate the data into the manuscript if the reviewer wishes.

Author response image 1.**DOI:**
http://dx.doi.org/10.7554/eLife.08648.016

Author response image 2.**DOI:**
http://dx.doi.org/10.7554/eLife.08648.017

*5) In contrast to schema in*
Figure 5*, AT1R was shown to be activated in a ligand-independent manner (Cook JR et al. JCI 2014). The authors have not formally addressed angiotensin-2 dependency in the AT1R-activation.*

The paper by Cook et al. tackled the challenging topic of elucidating the mechanistic basis of dilated cardiomyopathy (DCM) in Marfan syndrome mice, using genetic, pharmacological and embryological manipulations. They used a different mouse model of Marfan syndrome, namely the *Fbn1*^MgR/MgR^ model. This line is homozygous for a hypomorphic allele of *Fbn1*. This differs from the *Fbn1*^C1039G/+^ mouse model we used, which is heterozygous for the most common class of missense mutation causing Marfan syndrome in people. While this may have some relevance, it is perhaps more important that the Cook paper did not address the phenotypic consequences of angiotensinogen gene knockout in the aorta – the sole focus of this manuscript. It is also notable that the critical figure in their paper that assesses the phenotypic consequence of deletion of the angiotensinogen gene (*Agt*^*-/-*^ ; Figure 4) does not provide data for contemporaneous untreated Marfan mice – an essential control, but rather the angiotensinogen knockout Marfan mice are only compared to WT animals. Furthermore, no heart weight to body weight data or signaling effects are provided for the *Agt*^*-/-*^ animals. The only conclusion that can be reached from this aspect of the study is that Marfan mice lacking angiotensinogen have some degree of left ventricular dilatation and dysfunction, when compared to WT mice. This falls far short of suggesting that all cardiovascular aspects of Marfan syndrome relate to an AT1R-dependent but AngII-independent mechanism.

We have previously shown that elimination of the AT2R in Marfan mice reduces the therapeutic efficacy of AT1R blockade with losartan in these animals, and that this correlated with a lack of rescue of Erk1/2 activation in losartan-treated AT2R-knockout Marfan mice. This suggests that the protective effect of AT1R blockade in Marfan syndrome is, at least in part, dependent on AT2R signaling. Given that AngII is the major ligand for the AT2R, this suggests that AngII signaling (via the AT2R) plays a significant role in driving aortic disease pathology in Marfan mice. Given that ARBs work predominantly (if not exclusively) as a competitive inhibitor of AT1R, these data suggest that the therapeutic effect of ARBs in the aorta relates to displacement of AngII and shunting of signaling through AT2R. It seems notable that no mechanism beyond a vague reference to stretch and integrin signaling was offered by the Cook paper to explain ligand-independent AT1R activation in the heart. While we have not eliminated the angiotensinogen gene, this would seem a reasonable avenue for future work, but falls outside the scope of the current work focusing on the effect of calcium channel blockers in the aortic wall of Marfan syndrome mice.

Reviewer #2 (Additional data files and statistical comments):

*Appropriate statistical analyses (other than* t*-test) should be performed for the comparison of samples more than 3.*

The data has now been analyzed using two-way ANOVA for continuous data (echocardiography and Western blot analyses), and Kruskal-Wallis ANOVA for categorical data (i.e. aortic architecture analyses).

Reviewer #3:

*1) Doses of amlodipine, verapamil, and hydralazine (in mg/kg/d) are much higher than those used in humans: ∼100-fold (the lower amlodipine dose is still 25-fold), ∼40-fold, and ∼5-fold, respectively. Authors must comment on the relevance of these murine doses to human therapeutics. Off-target effects seem likely. Relevant to this,*
Figures 1 and 2
*and 1-9 show that this dose of amlodipine increases aortic growth in wild-type mice by several-fold. Similarly verapamil increases aortic growth in wild-type mice by ∼10-fold. This is highly unlikely to apply to humans taking amlodipine and verapamil; why should the results in MFS mice be any more translatable to humans taking these drugs?*

Rodents are rapid metabolizers of drugs compared to humans, and so the half-life of a drug tends to be shorter in rodents. This is especially true of drugs whose half-life is influenced by volume of distribution, as is the case for amlodipine. Hence the half-life of amlodipine in rodents has been shown to be is one-third to one-tenth as long as humans (Stopher et al., 1988), necessitating the use of a significantly higher dose of the drug in rodent studies. Prior work in mouse models of cardiovascular disease have used doses of amlodipine ranging from 1-10mg/kg/day, with the greatest therapeutic effect being seen at the highest dose (Wang et al., 1997). We therefore chose to use a dose that maximized our chance of seeing a therapeutic effect from amlodipine, while simultaneously reducing blood pressure equally as much as losartan in our mouse model. Hence we chose to use a dose of 12mg/kg/day. As one can see from Figure 1—figure supplement 1, both losartan and amlodipine showed approximately a 25% reduction in systolic and diastolic blood pressure. For hydralazine, we found that we could achieve a full rescue of aortic root growth using a dose that was only ∼5 fold greater than that used in humans, and hence there seemed no need to use a higher dose.

Baseline growth of the ascending aorta is minimal in placebo-treated mice, and so any accentuation of growth appears to have a dramatic effect. Relative to its overall diameter (about 1.6mm), even growth of 0.12mm/2mo only represents a 6-7% increase in overall diameter in CCB-treated WT mice. Furthermore, it never progressed to a clinically relevant outcome such as death in WT mice. It is feasible that modest increases in ascending aortic size could occur in humans on CCBs, but may not be of sufficient magnitude to have led to detection by in-vivo imaging, or to any clinically relevant outcomes such as dissection or premature lethality. One might argue that our findings in wild type mice warrant further investigation in humans.

2) The figures should use consistent units and axes so results can be easily compared. Aortic growth should be in mm/2mo only. The scale for all aortic root graphs should be identical in all figures as should the scale for all ascending aorta graphs.

The Y-axis in all echocardiography graphs has been adjusted to growth in mm/2mo to allow direct comparisons.

3) In the subsection “Effect of Non-Dihydropyridine CCBs: Verapamil,” ascending aortic growth in verapamil-treated MFS mice did not “closely parallel” that in amlodipine-treated mice. It was much less (∼ 75% less). Maybe this is because the verapamil dose was only 40 times the typical human dose but the amlodipine dose was 100 times the typical human dose.

It was previously stated that the effects of verapamil “closely paralleled” the effects of amlodipine, since there was the same trend, namely a small effect in WT mice and a much greater accentuation of growth in Marfan mice, with the effect being greater in the ascending aorta than the aortic root. The reviewer is correct in highlighting that while the trend is the same, amlodipine had a much more pronounced effect, so the text has been updated to reflect this. Even the lower dose of amlodipine had a greater effect than verapamil, suggesting that amlodipine has a more deleterious effect than verapamil. While the 2 drugs target L-type calcium channels, verapamil is generally considered to be more cardioselective, with amlodipine having a greater tropism for aortic cells, which may explain why amlodipine has a more deleterious effect than verapamil on aortic growth.

*4) All of the western analyses for phosphorylated signaling molecules must be normalized to their unphosphorylated forms, not to β-actin. Also, in*
Figure 3*, pSmad3 is the upper not the lower band.*

The Western blot analyses have been amended to include data for total proteins, in addition to a standard loading control in the form of β-Actin. This information has also been added into the Methods section. Prior work found no difference when phosphorylated proteins from the aortas of WT or Marfan mice were quantified to total proteins or β-Actin (Holm 2011). Again, we find no significant difference in the conclusions to be drawn from the data when quantification of phosphorylated proteins is compared to their respective total proteins or β-Actin.

To make interpretation easier and to avoid confusion, the lower band on the pSmad3 blot has been removed.

*5) Ascending aortic growth is minimally variable within an experiment, but highly variable between experiments (compare*
Figures 1 and 3*). I have the same concern for aortic root growth (compare*
Figures 1 and 4*). Is there a good explanation?*

The multiple studies presented in the paper were conducted over a number of years. During this long time period, we cannot exclude a possible contribution from a number of variables that may have influenced the magnitude of aortic growth of the mice and/or their therapeutic response to the drugs, including genetic drift within the mouse colony, changes in the animal facility, and lot-to-lot variability in the relative potency of various experimental compounds. To accommodate for such variables, we only performed analyses comparing contemporaneous cohorts of mice, using littermates and large sample sizes in all studies.

*6)*
Figure 5
*seems at odds with earlier work from this group in which TGF-β is identified as the primary initiator of aortopathy in MFS mice (due to excessive activation associated with mutant fibrillin). Here, angiotensin-2 is portrayed as the primary initiator, with TGF-b activation downstream. Also, the figure suggests that aortopathy could occur independently of TGF-β. Please explain.*

The diagram has been modified to more accurately reflect our current thinking about signaling in Marfan mice. Abnormal or insufficient fibrillin-1 in Marfan syndrome leads to aberrant activation of TGFβ, which in turn leads to upregulation of both canonical (Smad) and non-canonical (ERK) TGFβ-dependent signaling cascades. AngII, via the AT1R, can accentuate this by upregulating TGFβ signaling, through increased expression of TGFβ ligands, receptors, and activators (Wolf, Ziyadeh and Stahl, 1999; Fukuda et al., 2000; Naito et al., 2004). The current model illustrates the cross-talk that is believed to occur between the angiotensin and TGFβ signaling cascades and that is critical to driving aortic disease pathogenesis in Marfan mice. This model highlights why treatment with either TGFβ neutralizing antibody or the AT1R blocker losartan can lead to therapeutic rescue in Marfan mice. Hence the current model does not contradict our prior work, but rather complements it, by identifying the mechanism of CCB-induced aortic aneurysm exacerbation, and further defining the downstream pathways that drive aortic disease pathogenesis in Marfan mice. It is known that infusion of supraphysiological levels exogenous AngII can induce aneurysm formation and progression in mice (the so-called AngII infusion aneurysm model). Hence the diagram accommodates the fact that this can occur.

7) The GenTac analyses seem to have a fundamental flaw. Because β-blockers are first-line therapy and CCBs would typically be given to patients who could not tolerate β-blockers, the observations here may be entirely due to lack of β blockade rather than use of CCBs. To avoid this, the analyses here must either be controlled for β-blocker use or, alternatively, a propensity-matching approach must be used.

This interpretation is directly at odds with our mouse model data. The animal work indicates that disease acceleration occurs due to the presence of a deleterious factor (i.e. CCB), not simply due to the absence of a protective agent (i.e. β-blocker). If the latter were the case, then one would expect CCB-treated Marfan mice to show the same growth rate as placebo-treated Marfan mice, which was not the case.

To further assess this, we controlled for β-blocker use in our GenTAC analysis and have now included this in the results (Table 1). From the data, one can see that controlling for β-blocker use did not fundamentally alter the conclusions of the study. The odds of aortic dissection and aortic surgery remained significantly increased in Marfan patients on CCBs, while the odds of aortic surgery remained significantly increased in patients with other forms of inherited aortic aneurysm on CCBs.

Reviewer #3 (Additional data files and statistical comments):

*1) Relevant to concern #1, the authors must objectively test whether the agents produce different results in wild-type versus MFS mice. This would be done by 2-way ANOVA, specifically the interaction term that tests whether (for example in*
Figure 1*) the effect of amlodipine on aortic growth depends on genotype. This is particularly important because the title reports a “gene by environment interaction” whereas no statistical support for a significant interaction is present in this manuscript.*

As requested, two-way ANOVA have now been performed for the continuous echocardiography and Western blot analyses. Indeed the revised data show that the interaction between genotype and drug treatment is significant for both amlodipine and verapamil, supporting the wording of the title and the conclusions of the manuscript.

*2) The statistics section reports only use of the 2-tailed* t*-test. The aortic architecture data, however, is obtained with a categorical not a continuous scale and should be analyzed with nonparametric tests (rank-sum and Kruskal-Wallis ANOVA). In addition,* t*-tests are used for multi-group experiments. This is not proper.*

As requested, Kruskal-Wallis ANOVA have been performed for all of the categorical aortic architecture data.

3) Figure legends should all report the number of animals per group.

The figure legends have been amended to include this information.

4) With regards Figure 3—figure supplement 1, did RDEA have any effect on pSmad3?

RDEA119 was previously shown to have no effect on Smad signaling in the aortas of placebo-treated Marfan mice (14). To further assess this, we analyzed Smad3 activation in amlodipine-treated Marfan mice (Figure 3), and again found that it had no significant effect.